# Hard-Decision Coded Modulation for High-Throughput Short-Reach Optical Interconnect

**DOI:** 10.3390/e22040400

**Published:** 2020-03-31

**Authors:** Bin Chen, Yi Lei, Gabriele Liga, Chigo Okonkwo, Alex Alvarado

**Affiliations:** 1School of Computer and Information Engineering, Hefei University of Technology, Hefei 230009, China; bin.chen@hfut.edu.cn; 2Information and Communication Theory Lab, Signal Processing Systems Group, Department of Electrical Engineering, Eindhoven University of Technology, 5600 MB Eindhoven, The Netherlands; g.liga@tue.nl (G.L.); a.alvarado@tue.nl (A.A.); 3High Capacity Optical Transmission Laboratory, Electro-Optical Communication Group, Department of Electrical Engineering, Eindhoven University of Technology, 5600 MB Eindhoven, The Netherlands; c.m.okonkwo@tue.nl

**Keywords:** coded modulation, error correcting codes, hard decision decoding, geometric shaping, staircase codes, optical fiber communications

## Abstract

Coded modulation (CM), a combination of forward error correction (FEC) and high order modulation formats, has become a key part of modern optical communication systems. Designing CM schemes with strict complexity requirements for optical communications (e.g., data center interconnects) is still challenging mainly because of the expected low latency, low overhead, and the stringent high data rate requirements. In this paper, we propose a CM scheme with bit-wise hard-decision FEC and geometric shaping. In particular, we propose to combine the recently introduced soft-aided bit-marking decoding algorithm for staircase codes (SCCs) with geometrically-shaped constellations. The main goal of this CM scheme is to jointly boost the coding gain and provide shaping gain, while keeping the complexity low. When compared to existing CM systems based on *M*-ary quadrature-amplitude modulation (*M*QAM, M=64,128,256) and conventional decoding of SCCs, the proposed scheme shows improvements of up to 0.83 dB at a bit-error rate of $10^{-6}$ in the additive white Gaussian noise channel. For a nonlinear optical fiber system, simulation results show up to 24% reach increase. In addition, the proposed CM scheme enables rate adaptivity in single-wavelength systems, offering six different data rates between 450 Gbit/s and 666 Gbit/s.

## 1. Introduction

Modern optical fiber communication systems have been revolutionized by digital coherent technology via techniques such as pulse shaping, digital signal processing (DSP), and forward error correction (FEC) [1]. In order to support the Internet’s exponential traffic growth and improve the transmission reach, the most natural way of achieving this goal is using high spectral efficiency (SE) modulation formats and powerful FEC techniques. The combination of modulation format and FEC is also known as coded modulation (CM) [2,3].

The purpose of CM is to increase the number of information bits per channel use (or information rate). High-order modulation formats can boost the information rates; however, there is often a gap between the performance of a practical CM scheme and the corresponding theoretical limit. The gap is due to the use of uniform signaling with equidistant signal constellation points and the suboptimality of pragmatic FEC schemes. In order to enhance the performance of CM systems and reduce the two losses above, advanced FEC decoding architectures [3,4,5,6] and signal shaping [7,8,9,10,11] have received considerable attention in the optical communication literature. The most popular approach is to use a binary code mapped nonbinary constellation in a so-called bit-interleaved coded modulation (BICM) structure. This approach naturally leads to a separate optimization of the FEC and the modulation format.

One of the approaches to enhance the performance of the CM system is to employ signal shaping, which mimics a capacity achieving distribution. Signal shaping can be broadly categorized into probabilistic shaping (PS) [8,9,12,13,14] and geometric shaping (GS) [15,16,17,18]. By employing the same number of signal dimensions, PS has superior achievable information rates (AIRs) performance improvement over *m*-QAM and flex-rate adjustment by adjusting probability for a finite number of constellation points with respect to GS (Section 4.2, [3]). However, its performance advantage over GS comes at the cost of high computational complexity. Usually, long codewords are required to achieve good performance, and error bursts after deshaping are also a problem. The inherent serialized processing also makes it challenging for parallelized implementation. These issues limit the application of PS in high-speed low-latency short-haul optical links.

In GS, the constellation points have the same probability of occurrence, but they are not equally-spaced in the corresponding geometric space. Initially, GS was designed without taking the mapping strategy into account. For instance, many GS constellations are obtained by sphere packing arguments [19], and some of those were then used in fiber optical communications, e.g., in [20,21]. Other approaches include for example iterative polar modulation with a Gaussian distribution maximizing the mutual information (MI) [22,23]. Recently, end-to-end learning to maximize MI has been investigated in [24], and also experimentally demonstrated in [25].

Mapping-independent optimizations do not work well for BICM [26]. For BICM, the (mapping-dependent) generalized mutual information (GMI) should be optimized instead of MI. The challenge is to jointly design the geometry of the constellation and the mapping strategy. This problem has attracted attention in both academia [17,27,28,29,30,31] and industry [32,33,34].

FEC design is also key to improve pragmatic CM schemes performance. In general, FEC comes in two flavors: hard-decision (HD) and soft-decision (SD) FEC. HD-FEC decoders use binary representations of bits, whereas SD-FEC decoders use more accurate representation of the bits’ reliabilities. This “soft information” is often represented using logarithmic likelihood ratios (LLRs). Modern SD-FEC codes such as low-density parity-check (LDPC) offer a signal-to-noise ratio (SNR) improvement of ~1–2 dB compared with HD-FEC codes of the same rate. In general, SD-FEC yields large coding gains, but poses implementation challenges in terms of complexity, delay, power consumption, and circuit area [35]. Therefore, for applications with strict latency and complexity requirements (e.g., short reach), HD-FEC codes are the preferred alternative.

Staircase codes (SCCs) [36,37] and product codes (PCs) [38] are two popular families of codes that give large coding gains and allow low-complexity HD-FEC decoding structures. At present, PCs have been adopted by the optical submarine standard [39] and SCCs are part of the 400ZR Implementation Agreement (as an outer code) in the Optical Internetworking Forum [40]. PC and SCC decoders have been implemented and evaluated using synthesized gate netlists in a 28 nm in very-large-scale integration system, reaching more than 1 Tb/s information throughput with energy efficiency of ~2 pJ/bit only [41]. Closing the gap between the performance of HD-FEC and SD-FEC is an active research area. One idea for improving decoding is to use pure soft-input/soft-output decoder for a HD-FEC structure within all iterations [42], another idea is to use soft-aided algorithms with hard-input/hard-output decoder via scaled reliabilities [43,44,45], soft-aided bit-marking (SABM) [46,47], or a combination of the two [48].

In this paper, we propose a geometrically-shaped soft-aided SCC (GS-SA-SCC) coded modulation scheme. In particular, similar to the works in [17,31], we optimize the constellations and mapping strategy in terms of GMI. We then apply the SABM decoding algorithm for SCCs to yield additional coding gains. We show that SCCs with SABM decoding and GS provide gains up to 0.83 dB with respect to the baseline scheme with equidistant QAM and standard decoding for SCCs. Furthermore, we evaluate the proposed CM scheme in both single-span and multi-span wavelength-division multiplexing (WDM) optical fiber communication system, which covers a wide range of distances up to 950 km and line rates up to 666 Gbit/s. By evaluating the AIRs and post-FEC bit error rate (BER) results, we show that the proposed GS-SA-SCC CM scheme achieves up to 24% reach increases with respect to the system using equidistant QAM and standard decoding for SCCs for multiple line rates between 450 Gbit/s and 666 Gbit/s.

The paper is structured as follows. In Section 2, we discuss general aspects of AIRs, GS, and SCCs with SABM decoding as well as the proposed GS-SA-SCC CM scheme. In Section 3, we present and analysis the advanced CM scheme with geometrically-shaped constellations and SABM for SCCs over additive white Gaussian noise (AWGN) channel. The simulation results of optical CM transmission system for both single-span system and multi-span systems are further presented in Section 4. Finally, we conclude this paper in Section 5.

## 2. AIRs, CM, and Proposed Design

### 2.1. Achievable Information Rates (AIRs)

Achievable Information Rates (AIRs) have emerged as practical tools to design optical fiber communication systems, i.e., to design modulation formats and predict the performance of FEC [49,50,51]. The channel MI represents an upper limit on the AIRs for a given channel when a given modulation format is used along with an information-theoretic optimum receiver. However, the implementation of such a decoder is in general avoided for practical reasons. A widespread choice for performance prediction of (binary) SD-FEC is the GMI, which is suitable for bit-wise (BW) decoder (i.e., a standard BICM receiver) [3,50].

Throughout this paper, random (row) vectors are denoted by X and their corresponding realizations are denoted by x. Matrices are denoted by X. The transmitted bits are assumed to be independent and uniformly distributed, which implies uniform symbols X (no PS is used). The receiver assumes a memoryless channel and also uses a bit-wise decoder (i.e., a standard BICM receiver). Under the assumptions above, the AIRs for SD-FEC and BW decoder is GMI
(1)$$I_{\mathrm{SD}}=\sum_{i=1}^{m}I\left(B_{i};Y\right)=\sum_{i=1}^{m}E\left[\log_{2}\frac{f_{Y|B_{i}}\left(Y|B_{i}\right)}{f_{Y}\left(Y\right)}\right]$$
where the bits mapped to the channel input X are represented by the random variables $B=[B_{1},B_{2},\ldots ,B_{m}]$, $m=\log_{2}M$ is the number of bits per constellation point, Y represents channel output, $f_{Y|B_{i}}\left(Y|B_{i}\right)$ is the channel law and $I(B_{i};Y)$ is the MI between the bits and the symbols.

As for HD-FEC schemes, we neglect the bit position asymmetry by assuming an ideal interleaver between BW demapper and HD-FEC decoder. Under the assumptions above, the AIRs are given by [52]
(2)$$I_{\mathrm{HD}}=\sum_{i=1}^{m}I\left(B_{i};{\hat{B}}_{i}\right)≜m\left(1+p\log_{2}\left(p\right)+\left(1-p\right)\log_{2}\left(1-p\right)\right)$$
where ${\hat{B}}_{i}$ represents the estimated bit after demapper and *p* corresponds to the average pre-FEC bit error rate (BER) across the *m* bit positions. Note that an average binary symmetric channel capacity might be a pessimistic AIR for generic HD-BW decoders. However, it is a more suitable AIR for HD-FEC decoders that disregard bit position asymmetry.

Even though AIRs can be used to predict key metrics of communication systems, a gap between the AIR and the corresponding performance in practice is inevitable. In order to achieve a higher AIR, these penalties should be avoided. In the following, three SNR gaps, including the gap between HD-FEC and SD-FEC, shaping gap, and practical coding gap, are shown via an example.

**Example** **1.**
*AIRs in Equations (Equation 1) and (Equation 2) for 64QAM are shown in Figure 1 (solid red and solid blue curves, respectively). These results show the theoretical gap (blue shaded area) of 1–2 dB. The SNR gap caused by using HD-FEC instead of SD-FEC is relatively small in the high-SNR regime (≈1 dB) with respect to the low-SNR regime (≈2 dB). Figure 1 also shows the gap (gray shaded area) between the Shannon capacity and ideal SD-FEC, which can be reduced by using a high-order and shaped modulation with an optimal distribution.*

*AIR analysis relies on an ideal assumption that there exists a FEC code with infinitely long codewords, which achieves an arbitrarily low BER at a code rate R=AIR. However, codes with a finite block length and a complexity-limited implementation will lead to a loss with respect to the AIR prediction. Here, we show the performance of LDPC codes using soft-decision decoding and SCC codes using hard-decision decoding (without any soft information) to achieve a post-FEC BER below $10^{-6}$. A simple concatenated outer code, i.e., BCH, can bring the post-SCC BER down to $10^{-15}$. The results are shown with markers in Figure 1. Binary LDPC codes are simulated with code rates R∈{3/5,2/3,3/4,5/6,8/9} from the DVB-S2 standard, with a block length of N= 64,800 bits, and 50 decoding iterations. SCCs are simulated with higher code rates R∈{0.83,0.85,0.90,0.92} and block size w∈{114,126,192,252}, where w follows the notation in ([47], Figure 1b). The component codes we chose for these SCCs are the most popular ones, i.e., binary Bose–Chaudhuri–Hocquenghem (BCH) codes but extended with 1 extra parity bit. The error-correcting capability of these BCH codes are 2. Decoding is done using iterative bounded-distance decoding (BDD) with a window length of 8 and 7 iterations per window. More details about this are given in Section 2.2.*

*Figure 1 shows that both LDPC and SCC have approximately 1 dB gap to their corresponding AIRs. It is also reported in [53,54,55] (SC-LDPC) that hardware-implementable state-of-the-art SD-FEC codes also have approximately 0.5–1.5 dB penalty from the AIR because of implementation limitations such as power consumption and memory limitations. The gap between AIR and practical FEC is due to the subopimalty of the codes and can be decreased by increasing the code length or performing a more efficient decoding algorithm. Although AIRs give a good first indication of performance, post-FEC BER simulations are necessary to evaluate the performance of a given FEC.*


From the example above, we can see that there exist opportunities to enhance the achieved data rate in practice by improving the CM scheme to reduce the gaps. Considering the strict latency and complexity requirements, simple but powerful CM systems (e.g., CM with HD-FEC) are desirable. Optimizing the modulation format and improving the FEC performance is therefore of great importance.

### 2.2. Advanced CM Schemes

#### 2.2.1. Improved Modulation

GS constellations are generated by optimizing a cost function under certain constraints and typically for a given channel. The cost functions could be symbol error probability [56], MI [17,22,24], GMI [17,28], etc. Throughout this paper, we consider BICM, which is one of the most popular CM schemes due to its advantages of flexibility, simplicity, and parallelizability. The transmitted symbols X symbols with two real dimensions drawn uniformly from a discrete constellation X with cardinality $M=2^{m}=\left|X\right|$. The *j*th constellation point is denoted by $s_{j}=\left[s_{j,1},s_{j,2}\right]\in R^{2}$ with j=1,2,…,M. We use the M×2 matrix $S={\left[s_{1}^{T},s_{2}^{T},\ldots ,s_{M}^{T}\right]}^{T}$ to denote the 2D constellation. The *j*th constellation point $s_{j}$ is labeled by the length-*m* binary bit sequence $b_{j}=\left[b_{j,1},\ldots ,b_{j,m}\right]\in {\left\{0,1\right\}}^{m}$. The binary labeling matrix is denoted by a M×m matrix $B={\left[b_{1}^{T},b_{2}^{T},\ldots ,b_{M}^{T}\right]}^{T}$, which contains all unique length-*m* binary sequences. The 2D constellation and its binary labeling are fully determined by the pair of matrices {S,B}.

Without the loss of generality, we consider only one of the polarizations for the shaping optimization; however, we emphasize that all results in this paper is performed for 2D constellations on each polarization. Therefore, we can optimize the two-dimensional constellation coordinates and the corresponding labeling by maximizing the cost function. For BW decoders, the AIR-based optimization problem is to find a constellation $S^{*}$ and labeling $B^{*}$ for a given channel law $f_{Y|X}$ and an energy constraint, i.e.,
(3)$$\begin{array}{cc} \{S^{*},B^{*}\}= & \underset{{S,B:E[∥X∥}^{2}]\leq \sigma_{x}^{2}}{\mathrm{argmax}}G\left(S,B,f_{Y|X}\right) \end{array}$$
where $G(S,B,f_{Y|X})$ is equal to either $I_{\mathrm{SD}}$ for SD-FEC or $I_{\mathrm{HD}}$ for HD-FEC; $\sigma_{x}^{2}$ represents the transmitted power; and $S^{*}$ and $B^{*}$ indicate the optimal constellation and labeling, respectively. In this paper, average power constraint is considered in the optimization. Besides that, peak-to-average power ratio constraint is commonly used in optical fiber communication to maximize the signal generation performance for the systems with limited resolution of digital-to-analog converters and power budget of the links.

Note that the optimization problem is a single objective function (AIR maximization) with multiple parameters. Many optimization methods can be used to find the optimal solution. This includes for example pairwise optimization algorithm [16,27,57], genetic algorithms [58], and machine learning approaches [24,59,60,61,62]. It is well known that a constellation with Gray-mapping has the minimum loss between MI and GMI. However, due to the unavailability of Gray-mapping for most GS constellations, only jointly-optimized coordinates and labeling is capable of reducing the impact of non-Gray mapping constellation, and thus reduce the gap between GMI and MI.

The main advantage of GS over PS is its low implementation complexity, as only the mapping and demapping blocks of the transponder need to be modified by giving a look-up table. However, the GS constellations may result in more levels in both in-phase and quadrature branches. Therefore, the digital-to-analog converter/analog-to-digital converter may be required to be implemented with higher resolutions. In order to reduce the practical requirements, different constraints in Equation (Equation 3), such as symmetry, constant number of phase level, or amplitude, can also be used.

#### 2.2.2. Improved FEC

SCC is constructed with a sequence of binary matrices $C_{i}\in {\left\{0,1\right\}}^{w\times w}$, i=0,1,2,…, which are ordered like a staircase, as shown in ([47], Figure 1b). Following the notations in [47], let $\left(n_{c},k_{c},t\right)$ denote the parameters of BCH code C, where $n_{c}$ is the codeword length, $k_{c}$ is the information length, and *t* is the error-correcting capability. The code rate of SCC with component code of C is $R=2k_{c}/n_{c}-1$, while the size of $C_{i}$ is $w=n_{c}/2$. Standard decoding of SCCs is performed using a sliding window, wherein BDD is used to iteratively decode each component code. BDD is very simple, however, its error correcting capability is limited to $t=\lfloor \frac{d_{0}-1}{2}\rfloor$ errors, where $d_{0}$ is the minimum Hamming distance. Sometimes, BDD will erroneously decode a received sequence with more than *t* errors, a situation known as a *miscorrection*. Miscorrections are highly undesirable and are known to degrade the performance of iterative BDD by introducing additional errors into the iterative decoding process.

In order to identify miscorrections and address those sequences with larger than *t* errors, SABM decoder exploits the channel soft information (see the two red highlighted blocks in Figure 2). The SABM decoder calculates the absolute value of LLR (|λ|) to know about the reliability of each bit (a higher value of |λ| indicates a more reliable bit). According to the values of |λ|, the SABM decoder marks some highly reliable bits (HRBs) and highly unreliable bits (HUBs). If |λ| is larger than a threshold δ, the corresponding bit will be marked as a HRB. The selection of δ is discussed in [47] (Section 6). On the other hand, the $d_{0}-t-1$ bits with smallest |λ| values in each row of SCC block $C_{i}$ are marked as HUBs.

With the marked information, once the component BDD decoder has finished decoding, the SABM decoder will check its decoding status. Only when BDD succeeds and the output is not in conflict with HRBs and/or zero-syndrome component codewords, the output will be accepted by the SCC decoder. Otherwise, it will be marked as a miscorrection and the suggested bit flips will be rejected. If miscorrection happens, SABM flips the $d_{0}-t-w_{H}\left(e\right)$ most unreliable bits from the HUBs, where *e* is the error pattern detected by BDD and $w_{H}$ denotes Hamming weight. For BDD decoding failures, the decoder will also flip the most unreliable bit from the HUBs. The intuition is that this approach in some cases will make the new sequence close enough to the transmitted codeword, i.e., with only *t* errors (which can be handled by the component BDD decoder).

The main feature of the SABM algorithm is its low complexity, as explained in what follows. The SABM algorithm only requires modifications to the decoding structure of the last block of each decoding window. Furthermore, in the SABM algorithm, each component code needs to be decoded at most twice. The relative increased complexity around the optimum threshold δ is only ~4% at the post-FEC BER of $10^{-4}$ with respect to standard decoding for SCC, in terms of the number of additional calls to the component BDD decoder. Also, the algorithm is based on marking bits only, which simply requires to store very small part of the soft bits (LLRs). Finally, marked bits do not need to be tracked and updated during the iterative process either. The complexity of the SABM algorithm has been discussed in details in [47].

#### 2.2.3. GS-SA-SCC

In the paper, we propose a CM scheme via combining geometric shaping and soft-aided SCC (GS-SA-SCC). As shown in Figure 2, information bits are encoded by a staircase encoder and then mapped to symbol $x_{k}$ from an obtained GS constellation with labeling via (Equation 3) as matrices pair $\left\{S^{*},B^{*}\right\}$, where *k* is the discrete time index. The receiver for the GS-SA-SCC uses an HD-based demapper to estimate the bits. The HD-estimated bits ${\hat{c}}_{1,k},\ldots ,{\hat{c}}_{m,k}$ are then decoded by a staircase decoder implemented with SABM decoding. The LLR values $\lambda_{1,k},\ldots ,\lambda_{m,k}$ for the *m* bits per symbol are calculated as in [50] (Equation (25)), according to the channel information $y_{k}$ and also the transmitted modulation format. The marked information is denoted by ${\hat{q}}_{i,k},i=1,\ldots ,m$, which can be HRB, HUB, or unmarked.

Typically, GMI-optimized constellations translate into improved performance of the binary SD-FEC. However, it also has been shown that the GMI-optimized constellations could provide a considerable gain for a HD decoder [31,63]. In addition, the FEC we considered in this paper is a hybrid HD/SD FEC. Due to the use of a soft-aided decoding algorithm, it should in principle also benefit from the GMI-optimized constellations. The intuition behind the chosen performance metric is that for a hybrid HD/SD decoding, the more soft information is harvested in the FEC decoder, more potential gains can be achieved by using GMI-based optimization. For these reasons, we optimize the 2D constellation by maximizing $I_{\mathrm{SD}}$ instead of $I_{\mathrm{HD}}$ in (Equation 3). In the next sections, we will investigate how much improvement will be achieved by using the proposed GS-SA-SCC CM scheme.

## 3. Performance Over AWGN Channel

In this section, we show the performance over the most commonly used channel for optical communications, i.e., AWGN channel, which neglects the correlation caused by dispersion and nonlinear interference. Figure 2 shows a schematic diagram of the proposed CM communication system. The channel block in Figure 2 is modeled under a circularly symmetric Gaussian noise assumption, i.e., $p_{Y|X}$ is fully determined by the variance $\sigma_{z}^{2}$, which corresponds to the total noise power per one complex dimensions.

By solving the optimization problem in Equation (Equation 3), we obtained three groups of geometrically-shaped constellations with SEs of 6 bit/2D-sym, 7 bit/2D-sym, and 8 bit/2D-sym as follows.
For the geometrically-shaped 64-point modulation format (64-GS), we optimized the constellation and its labeling for SNR = 19 dB. The 64-GS constellation and left-MSB (most significant bit) bits mapping are plotted in Figure 3. The 64-GS format with $M=2^{m}=64$ points is labeled by m=6 bits, which are represented by $\left(b_{1},b_{2},b_{3},b_{4},b_{5},b_{6}\right)$ and the last two bits $\left(b_{5},b_{6}\right)$ determine the quadrants of the constellation point (marked in different color). The first four bits determine the one of the 16 points in each quadrant.For the geometrically-shaped 128-point modulation format (128-GS), we optimized the constellation and its labeling for SNR = 23 dB. The 128-GS constellation and bits mapping are plotted in Figure 4a.For the geometrically-shaped 256-point modulation format (256-GS), we optimized the constellation and its labeling for SNR = 25 dB. The 256-GS constellation and bits mapping are plotted in Figure 4b. The 256-GS format with $M=2^{m}=256$ points is labeled by m=8 bits, which are represented by $\left(b_{1},b_{2},\ldots ,b_{8}\right)$, and the two bits $\left(b_{4},b_{8}\right)$ determine the quadrants of the constellation point (marked in different color). The other six bits determine the one of 64 points in each quadrant.

For SCCs, two component codes, i.e., BCH codes (504,485,2) and (228,209,2), are considered. They correspond to the acceptable overhead range between 7% and 20% in optical fiber communications, which are the highest and lowest code rates investigated in Figure 1. These parameters are obtained by shortening the 1 bit-extended BCH code (512,493,2) by 8 and 284 bits, respectively. The extended bit is a parity bit and added at the end of standard BCH code (511,493,2). These two BCH codes result in SCC code rates R≈0.92 and R≈0.83, respectively, and SCC block size w=252 and w=114, respectively The decoding window size is 8, and the number of iterations is 7. The threshold δ for marking HRBs is 5. Note that the threshold here is equivalent to δ=10 in [47] due to the different definition of the SNR. Here, the SNR is defined as the energy per complex symbol over the noise variance over one complex dimension.

Figure 5 shows the post-FEC BER performance vs. SNR for SCC code rates of R=0.83 (circles) and 0.92 (triangles). Furthermore, we investigate three CM schemes with M={64,128,256}: *M*QAM with SCC standard decoding (solid lines), *M*-GS with SCC standard decoding (dashed lines), and *M*-GS with SABM decoding (dotted lines). It can be seen from Figure 5 that for different code rates, 64-GS can provide up to 0.29 dB gain compared to 64QAM. In addition, 64-GS with SABM decoding further increases the gain up to 0.55 dB at the BER of $10^{-6}$. For M=128 and M=256, the gains increase up to 0.83 dB and 0.63 dB, respectively Note that for constellations representing an odd number of bits, e.g., 128QAM, it is not possible to obtain a Gray mapping. Therefore, larger shaping gains of 128-GS are observed with respect to 128QAM.

In Figure 6, we depict the AIRs in Equation (Equation 2) for HD-FEC schemes with *M*QAM and *M*-GS. As can be seen, the GS constellations lead to a SNR gain in terms of SNR compared to uniform QAM in the case of multiple SEs. For a normalized AIR of 0.9 (diamond markers), the shaping gains of GS are 0.3 dB, 0.6 dB, and 0.35 dB for M={64,128,256} with respect to *M*QAM, respectively. Note that the AIRs of 64QAM and 256QAM in Equation (Equation 1) for SD-FEC schemes are also shown as baselines in Figure 6. The gaps between $I_{\mathrm{SD}}$ and $I_{\mathrm{HD}}$ indicate a potential performance region for hybrid HD/SD FEC schemes. Figure 6 also shows (with markers) the SNR required for different CM schemes to achieve a post-FEC BER of $10^{-6}$. We remark that such a SNR gain is due to a twofold advantage of GS-SA-SCC over conventional SCC with QAM: (i) the shaping gains, which comes from the optimized constellation matrices pair $\left\{S^{*},B^{*}\right\}$, and (ii) the improved decoding capability provided by the SABM decoding. For GS and SCC with code rate of 0.83, we observe the shaping gains of GS are 0.3 dB, 0.6 dB, and 0.35 dB, which is in excellent agreement with the prediction of AIR curves. Thanks to the processing of SABM, it enables an additional gain on top of GS. The overall (shaping plus coding) gains for M={64,128,256} with GS and SABM decoding are up to 0.55 dB, 0.83 dB and 0.63 dB with respect to *M*QAM with standard decoding, respectively. Note that GS-SA-SCC with 128-GS and code rate of 0.92 even achieved almost the same rate as $I_{\mathrm{HD}}$ with 128QAM (see green square marker and blue dotted curve in Figure 6). Comparing to the results in Figure 1, GS-SA-SCC with code rate of 0.83 reduced 0.55 dB of the ~2 dB gap to LDPC with very small added complexity. In general, the $I_{\mathrm{HD}}$ is the AIR of decoding system operating solely based on HD decoding, while the soft-aided decoders operate beyond this. In other words, the overall gains shown here are larger than $I_{\mathrm{HD}}$ prediction. An interesting research is to find the performance metric of the soft-aided decoders, which is left for future work.

## 4. Numerical Analysis for an Optical Transmission System

In this section, numerical results of optical fiber transmission are presented by considering the channel block in Figure 2 as a nonlinear optical channel. As shown in Figure 2, the simulated system consists of an optical link comprising either unrepeatered single standard single-mode fiber (SSMF) span $N_{sp}=1$ or multiple SSMF spans with erbium-doped fiber amplifier (EDFA). The investigated link lengths are the typical distances of data center interconnects (<200 km), metro, and regional links (200–1000 km). To support the targeted net data rates of >600 Gbit/s and >400 Gbit/s per wavelength for these two distance ranges above, two sets of constellations are investigated. The first set uses 256QAM and 256-GS for single span transmission, and second set uses 64QAM, 128QAM, 64-GS, and 128-GS for multi-span transmission. The fiber propagation is simulated through the split-step Fourier method numerical solution of the nonlinear Manakov equation, where the step size was uniformly set to 0.1 km. The maximum-reach gains are defined as the reach increase observed at fixed net data rate when switching from 64QAM, 128QAM, and 256QAM constellations to 64-GS, 128-GS, and 256-GS. The QAM or GS symbols are independently modulated over the two orthogonal polarization channels via polarization multiplexing. The simulation parameters are shown in Table 1.

### 4.1. Single-Span Fiber System Setup

We first simulated a single-span transmission link over SSMF. At the transmitter, the information bits of each polarization are independently encoded and the modulator maps the encoded bits either to 256QAM or to 256-GS symbols. After the CM encoder, the signal was modulated using a root raised cosine (RRC) filter. Dual-polarization 11×45 GBd WDM channels are generated with 50 GHz spacing, yielding 720 Gbit/s raw data rate per wavelength. The corresponding line rate per wavelength is 720·R, where *R* is the code rate of FEC code. For each polarization of each WDM channel, independent sequences of $2^{18}$ symbols were transmitted. An ideal coherent receiver is used for detection and analog-to-digital conversion. Chromatic dispersion is digitally compensated by a fast Fourier transform based filter, then the signal is matched filtered and downsampled at the Nyquist rate.

#### 4.1.1. AIR Results

Results after propagation over single-span SSMF are shown in Figure 7 as AIR vs. launch power per WDM channel ($P_{\mathrm{ch}}$). At the distance of 110 km, the GS gain, which is the difference (in AIR) between 64QAM and 64-GS constellation, is 0.17 bits/sym at the optimal launch power (2.5 dBm). At the distance of 120 km, the GS gain increases to 0.21 bits/sym at the optimal launch power (3.5 dBm).

#### 4.1.2. Post-FEC BER Results

In order to evaluate the reach increase for the proposed CM for single-span transmission, we implemented SCCs for the central WDM channel (number 6) at the optimum launch power, which is obtained from Figure 7. Figure 8 shows the post-FEC BER performance for 256QAM and 256-GS modulation formats at two different line rates R=233/252≈0.925 and R=95/114≈0.833. For 256-GS and SCC with standard decoding at 666 Gbit/s and 600 Gbit/s, we observe a shaping gain of >2 km compared to 256QAM. The transmission reach extension of 256-GS and SCC with SABM decoding decoding is measured to be 4 km (3.7%).

### 4.2. Multi-Span Fiber System Setup

In the considered multi-span transmission link scenario, each SSMF span of length 80 km was followed by an EDFA compensating exactly for the link loss (16 dB) and with a noise figure of 5 dB. The receiver DSP algorithms utilised are identical to the ones described in single-span transmission scenario. $I_{\mathrm{HD}}$ of four different modulation formats are calculated to predict the shaping gain and finding the optimal launch power. In addition, pre-FEC BER and post-FEC BER are measured to verify the performance of the CM scheme.

#### 4.2.1. AIR Results

In Figure 9, two sets of results are shown for AIRs versus launch power over two multi-span transmission link, respectively. The first set (see Figure 9a) show the AIR for SE of 14 bits/sym, i.e., 128QAM and 128-GS, over 4×80 km SSMF and 6×80 km SSMF. Around the optimum launch power regime where linear and nonlinear propagation effects are comparable, 128-GS modulation gives 0.24 bit/sym and 0.5 bit/sym AIR gains with respect to 64QAM for over 4×80 km SSMF and 6×80 km SSMF, respectively

The second set (see Figure 9b) show the AIRs for SE of 12 bits/sym, i.e., 64QAM and 64-GS, over 10×80 km and SSMF 15×80 km SSMF. At the optimum launch power, 64-GS modulation gives 0.17 bit/sym and 0.19 bit/sym AIR gains with respect to 256QAM for over 10×80 km SSMF and 15×80 km SSMF, respectively Note that for higher launch power regime where nonlinearity is dominant, performance improvements are still remaining.

#### 4.2.2. Post-FEC BER Results

In order to evaluate the reach increase for the proposed CM for multi-span transmission, we implemented SCCs for multiple transmission distances and compared the post-SCC BER of *M*QAM and *M*-GS modulation formats (M∈{64,128}). Two code rates, $R_{s}^{1}=233/252$ and $R_{s}^{2}=95/114$, are used for SCCs. Therefore, the line rates of 450 Gbit/s, 500 Gbit/s, 525 Gbit/s, and 583 Gbit/s can be obtained. In addition, we concatenated LDPC codes with code rate $R_{l}=0.9$ (as inner code) and a SCC with code rate $R_{s}^{1}$ (as outer code) as a SD-FEC CM scheme baseline. The LDPC codes are the ones from the DVB-S2 standard, with a block length of N= 64,800 bits, and 50 decoding iterations. The decoded bits from the LDPC decoder are used as input to the staircase decoder. The overall concatenated code rate is $R=R_{s}^{1}\cdot R_{l}\approx R_{s}^{2}$, which is the code rate as SCC with $R_{s}^{2}=95/114$.

In Figure 10, post-FEC BER performance of central WDM channel at the optimum launch power is shown for *M*QAM and *M*-GS modulation formats (M∈{64,128}) at different line rates. The transmission reach extension of 64-GS with HD-FEC (SCC) at 500 Gbit/s and 450 Gbit/s are measured to be 80 km (13%) and 140 km (17%), respectively. Considering the higher complexity of the CM system combining 64QAM and SD-FEC (LDPC+SCC) as a baseline at the date rate of 450 Gbit/s, we observe only a gain of 75 km compared to the proposed scheme with 64-GS and SABM decoding. Therefore, the combination of HD-FEC and GS constellation offers a good complexity–gain trade-off. For a transmission range between 300 km and 500 km, the reach extension of 128-GS with HD-FEC (SCC) at 583 Gbit/s and 525 Gbit/s, are measured to be 70 km (24%) and 80 km (20%), respectively.

## 5. Discussion

The performance improvements of the proposed GS-SA-SCC CM schemes are summarized in Table 2, when compared to the existing CM schemes with standard SCC decoding and QAM modulations. Due to the greater sensitivity to the fiber nonlinear effect, the performance degradation of the system with the higher order modulation format is more serious than that of system with the lower order modulation format. Therefore, the observed distance increase is not significant as the SNR gain in the AWGN channel for constellation with 256 points. However, we can see that the proposed CM scheme can cover the distance from 100 km to 1000 km and provide a net data rate of up to 666 Gbit/s, which are suitable for short-reach DCI and metro links. One of the advantages of the proposed CM schemes is the 50 GHz grid compatibility, which is very common in most of optical network. For future the transponder with a 75 GHz fixed grid, these CM scheme could be also extended to beyond 1 Tbps without significant modification.

## 6. Conclusions

In this paper, a coded modulation scheme based on geometric shaping and soft-aided staircase code (GS-SA-SCC) was proposed and studied. The GS-SA-SCC coded modulation scheme is based on simple modification of hard-decision staircase decoder with the help marking bits and optimized constellations with corresponding labeling. The analysis was performed both in terms of achievable information rates and post-FEC BER. Up to 0.83 dB gains compared to standard SCC decoding with conventional QAM were obtained with a low added complexity. In this paper, we showed that multiple line rates can be obtained by using three geometrically-shaped constellations and two code rates. By combining these constellations with two HD-FEC codes, six line rates between 450 Gbps and 666 Gbps can be obtained. Furthermore, reach increases of up to 24% for a wide range of distances (from 100 km to 950 km) for both single-span and multi-span systems were reported. We believe that the proposed GS-SA-SCC scheme is a promising candidate for next generation data center interconect compatible coherent 800 Gbps and 1 Tbps system with higher baud rate, while providing considerable reach increase.

## Figures and Tables

**Figure 1 entropy-22-00400-f001:**
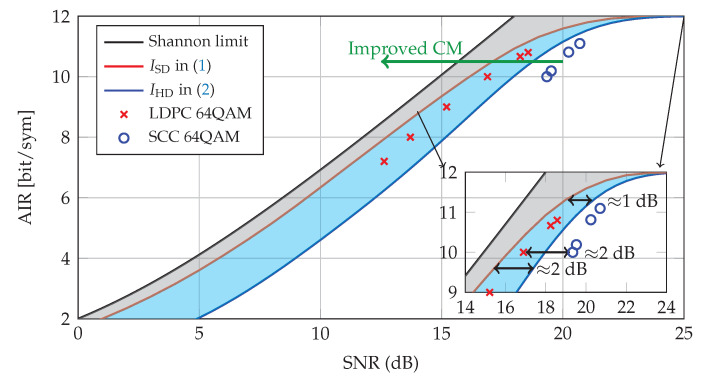
Signal-to-noise ratio (SNR) vs. achievable information rates (AIRs) for soft-decision forward error correction (SD-FEC) and hard-decision forward error correction (HD-FEC) with dual-polarization 64QAM. The results obtained with low-density parity-check (LDPC) codes and staircase codes (SCC) are shown with markers.

**Figure 2 entropy-22-00400-f002:**
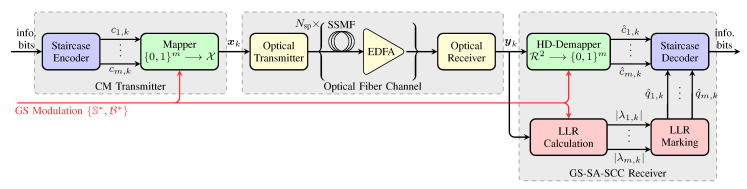
Optical fiber communication system model under consideration. The optical fiber channel is modeled using a channel with an optical fiber link comprising multiple spans ($N_{\mathrm{sp}}$ is the number of spans). Each span is followed by an EDFA. The highlighted green and red blocks are used to improve the performance of coded modulation in this paper.

**Figure 3 entropy-22-00400-f003:**
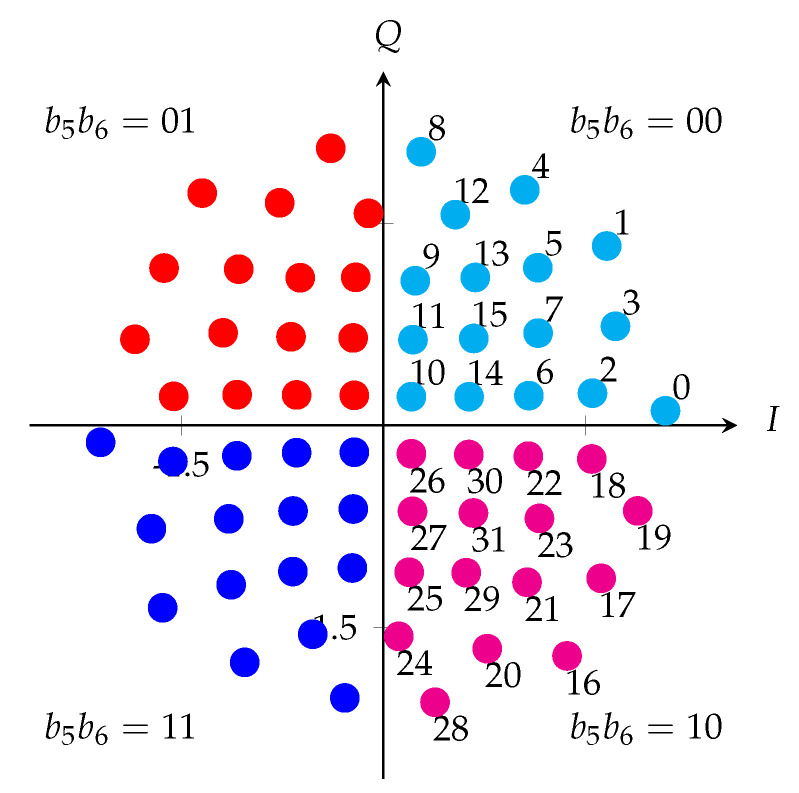
Constellation and labeling of 64-GS optimized for SNR = 19 dB with left-MSB bits mapping. The symbols in the same color are determined by the last two bits $b_{5}$ and $b_{6}$.

**Figure 4 entropy-22-00400-f004:**
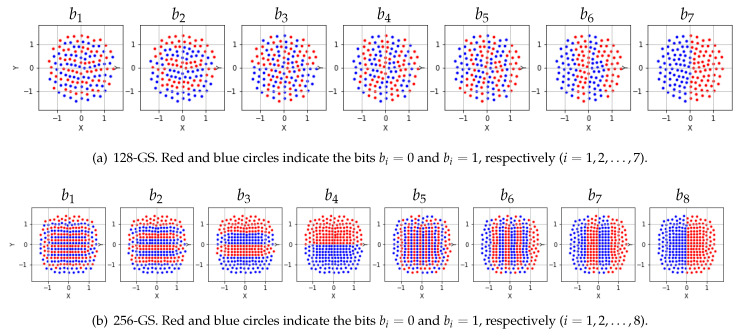
Constellation and labeling of 128-GS and 256-GS optimized for SNR = 23 dB and SNR = 25 dB, respectively.

**Figure 5 entropy-22-00400-f005:**
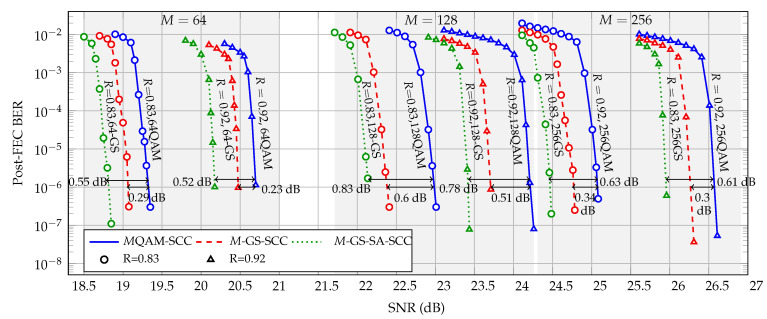
Additive white Gaussian noise (AWGN) performance: SNR vs. bit error rate (BER) for different modulation formats and SCC decoding algorithms.

**Figure 6 entropy-22-00400-f006:**
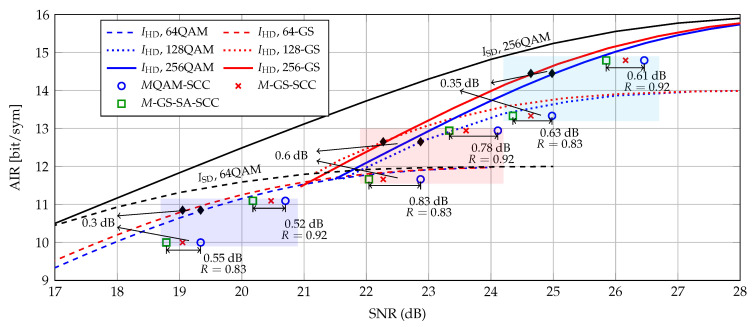
AIRs and operational CM rates vs. SNR for post-SCC BER < 1 × 10^−6^.

**Figure 7 entropy-22-00400-f007:**
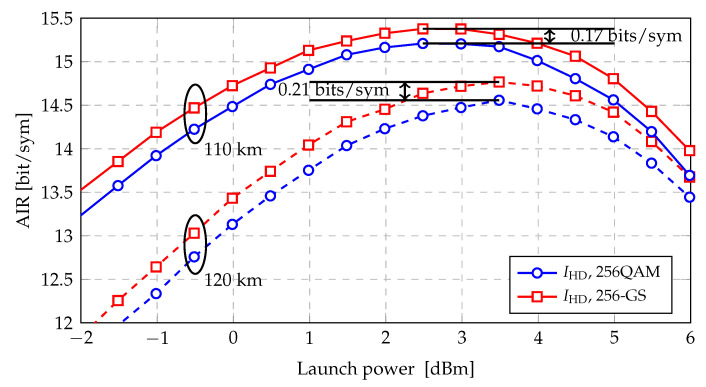
AIRs as function of the transmitted power for different modulation formats over single-span 110 km and 120 km SSMF.

**Figure 8 entropy-22-00400-f008:**
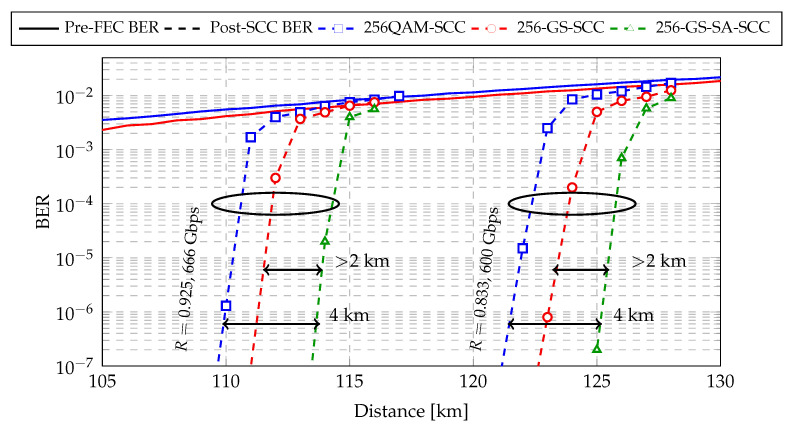
Post-FEC BER as a function of the transmission distance for different modulation formats and FEC codes over single-span transmission link.

**Figure 9 entropy-22-00400-f009:**
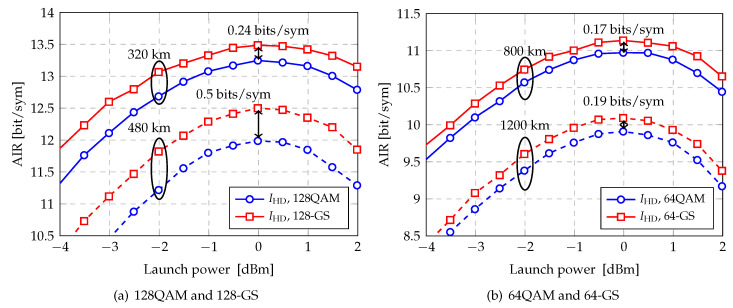
AIRs as function of the transmitted power for different modulation formats over multi-span SSMF.

**Figure 10 entropy-22-00400-f010:**
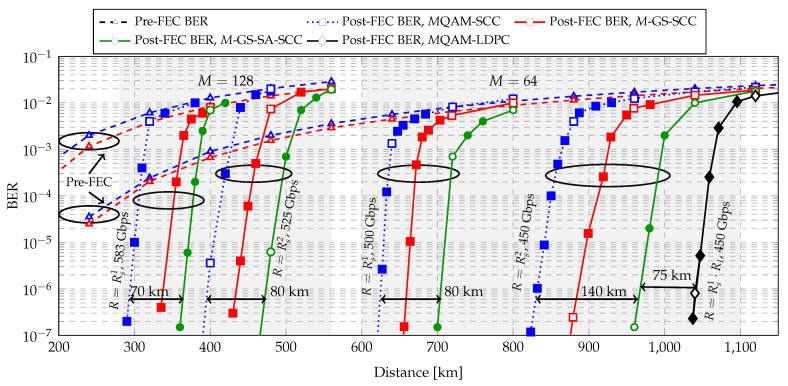
Post-FEC BER as a function of the transmission distance for *M*QAM and *M*-GS modulation formats (M∈{64,128}) and FEC codes. White markers indicate the BERs after the EDFAs. Filled markers show BER between two amplifiers (obtained by noise loading). SCC rates are $R_{s}^{1}=0.925$ and $R_{s}^{2}=0.833$. LDPC rate is $R_{l}=0.9$.

**Table 1 entropy-22-00400-t001:** Simulation parameters of the simulated optical link.

Parameter Name	Value
WDM Channels	11
Symbol rate	45 GBd
Root-raised-cosine roll-off factor	1%
Channel frequency spacing	50GHz
Center wavelength	1550 nm
Attenuation	0.2 dB ${\mathrm{km}}^{-1}$
Dispersion parameter	17 ps ${\mathrm{nm}}^{-1}{\mathrm{km}}^{-1}$
Nonlinearity parameter	1.2 $W^{-1}{\mathrm{km}}^{-1}$
EDFA noise figure	5 dB
Fiber span length (Multi-span)	80 km
**Modulation Format**	
Single-span system	256QAM/256-GS
Multi-span system	128QAM/128-GS
	64QAM/64-GS
**FEC**	
SCC	Standard/SABM decoding

**Table 2 entropy-22-00400-t002:** The summarized performance improvements of the proposed GS-SA-SCC CM schemes against the existing CM schemes with standard SCC decoding and QAM modulations.

Modulation format	256-GS		128-GS		64-GS	
Code rate	0.925	0.833	0.925	0.833	0.925	0.833
AWGN Channel						
SNR Gain (dB)	0.61	0.63	0.78	0.83	0.52	0.55
Optical Fiber Channel						
Distance increase	3.7%	3.3%	24%	20%	13%	17%
Distance (km)	<114	<125	<350	<450	<700	<950
Info. rate (Gbit/s)	666	600	583	525	500	450

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
