# Peer review of "Hard-Decision Coded Modulation for High-Throughput Short-Reach Optical Interconnect"

_entropy, 2020, doi:10.3390/e22040400_

Round 1

Reviewer 1 Report

This paper presents a  new coded modulation scheme based on a combination of a soft-aided bit-marking decoding algorithm for staircase codes (SCCs) with  geometrically-shaped constellations. The resulting scheme is the geometric shaping and soft-aided staircase code (GS-SA-SCC). Te results show an improvement over existing approaches.

Although quite marginal from an originality point of view, the paper presents certain features that could be of interest to the scientific community. Nevertheless, the following contents need to be improved:

Line 21 to 23 do not present a clear idea, please check the writing.

Delete commas after the equations

In line 119, it is mention Figure 1(b), but in Figure 1 it is not presented who is Fig. b.

In line 125, an extra symbol appears.

In section 2.2.1, it is mention that BICM was considered for its popularity, please add more advantages.  

Adding a discussion section would be desirable to better understand the results. 

Add tables to better understand the results and advantages over existing approaches.

Author Response

We thank the reviewers for their careful reading and valuable comments, which helped us to improve the paper. We did our best to address all the comments and suggestions, Please refer to the attached Response Letter.

Reviewer 2 Report

The paper is well written. I recommend publication.

Author Response

We thank the reviewers for their careful reading and valuable comments, which helped us to improve the paper. We did our best to address all the comments and suggestions, please refer to the attached Response Letter.
